

# Investigating the effect of the reality gap on the human psychophysiological state in the context of human-swarm interaction

Gaëtan Podevijn[1], Rehan O'Grady[1], Carole Fantini-Hauwel[2] and Marco Dorigo[1]

[1] IRIDIA, Université Libre de Bruxelles, Belgium
[2] Research Center of Clinical Psychology, Psychopathology and Psychosomatic, Université Libre de Bruxelles, Belgium

## ABSTRACT

The reality gap is the discrepancy between simulation and reality—the same behavioural algorithm results in different robot swarm behaviours in simulation and in reality (with real robots). In this paper, we study the effect of the reality gap on the psychophysiological reactions of humans interacting with a robot swarm. We compare the psychophysiological reactions of 28 participants interacting with a simulated robot swarm and with a real (non-simulated) robot swarm. Our results show that a real robot swarm provokes stronger reactions in our participants than a simulated robot swarm. We also investigate how to mitigate the effect of the reality gap (i.e., how to diminish the difference in the psychophysiological reactions between reality and simulation) by comparing psychophysiological reactions in simulation displayed on a computer screen and psychophysiological reactions in simulation displayed in virtual reality. Our results show that our participants tend to have stronger psychophysiological reactions in simulation displayed in virtual reality (suggesting a potential way of diminishing the effect of the reality gap).

## INTRODUCTION

In a near future, swarms of autonomous robots are likely to be part of our daily life. Whether swarms of robots will be used for high-risk tasks (e.g., search and rescue, demining) or for laborious tasks (e.g., harvesting, environment cleaning, grass mowing) (*Dorigo, Birattari & Brambilla, 2014*, *Dorigo et al., 2013*), it will be vital for humans to interact with these robot swarms (e.g., supervise, issue commands or receive feedback).

Recently, human-swarm interaction has become an active field of research. More and more, researchers in human-swarm interaction validate their work by performing user studies (i.e., group of human participants performing an experiment of human-swarm interaction). However, a large majority of the existing user studies are performed exclusively in simulation, with human operators interacting with simulated robots on a computer screen (e.g., *Bashyal & Venayagamoorthy, 2008*; *Nunnally et al., 2012*; *De la Croix & Egerstedt, 2012*; *Walker et al., 2012*; *Kolling et al., 2013*; *Walker et al., 2013*; *Pendleton & Goodrich, 2013*; *Nagavalli et al., 2015*).

Corresponding author
Gaëtan Podevijn, gpodevij@ulb.ac.be

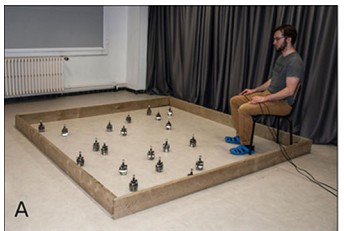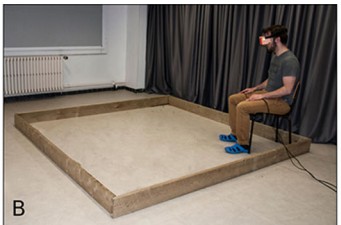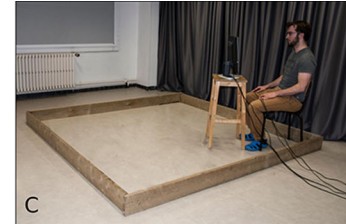

**Figure 1 Example of an experiment.** (A) A participant interacts with a swarm made up of 20 real robots. (B) A participant is attached to a virtual reality head set and interacts with a simulated swarm of 20 robots. (C) A participant interacts with a simulated swarm of 20 robots displayed on a computer screen. The participant shown in this figure is the first author of this paper and did not take part in the experiment. The pictures shown in this figure were taken for illustration purpose.

Simulation is a convenient choice for swarm roboticists, as it allows experimental conditions to be replicated perfectly in different experimental runs. Even more importantly, gathering enough real robots to make a meaningful swarm is often prohibitively expensive in terms of both money and time. However, conducting user studies in simulation suffers from a potentially fundamental problem—the inherent discrepancy between simulation and the reality (henceforth referred to as the reality gap).

In this paper, we study the effect of the reality gap on human psychology. Understanding the psychological impact of any interactive system (be it human-computer interaction, human-robot interaction or human-swarm interaction) on its human operator is clearly essential to the development of an effective interactive system (*Carroll, 1997*). To date, it is not yet clear what the effect of the reality gap is on human psychology in human-swarm interaction studies. Our goal is to study this effect.

We present an experiment in which humans interact with a simulated robot swarm displayed on a computer screen, with a simulated robot swarm displayed in virtual reality (within a virtual reality headset) and with a real (i.e., non-simulated) robot swarm (see Fig. 1). In our experimental setup, our goal was to produce results that were as objective as possible. To this end, we firstly recorded psychological impact using psychophysiological measures (e.g., heart-rate, skin conductance), which are considered more objective than purely questionnaire-based methods (*Bethel et al., 2007*). Secondly, we made purely passive the interaction of our human operators with the robot swarm. In this purely passive interaction, our participants do not issue any commands to, nor receive any feedback from the robot swarm. Finally, we decided that our participants would interact with a robot swarm executing a simple random walk behaviour (compared to a more complex foraging behaviour, for instance). These two choices allow us to isolate the reality gap effect. The passive interaction reduces the risk that psychophysiological reactions to the interaction interface (e.g., joystick, keyboard, voice commands) would be the strongest measurable reaction, drowning out the difference in reaction to the reality gap. The choice of a simple random walk behaviour reduces the risk that any psychophysiological reactions are caused by reactions to artefacts of a complex swarm robotics behaviour.

Our results show that our participants have stronger psychophysiological reactions when they interact with a real robot swarm than when they interact with a simulated robot

swarm (either displayed on a computer screen or in a virtual reality headset). Our results also show that our participants reported a stronger level of psychological arousal when they interacted with a robot swarm simulated in a virtual reality headset than when they interacted with a robot swarm simulated on a computer screen (suggesting that virtual reality is a technology that could potentially mitigate the effect of the reality gap in human-swarm interaction user studies). We believe the results we present here should have a significant impact on best-practices for the future human-swarm interaction design and test methodologies.

## RELATED LITERATURE

Human-swarm interaction, the field of research that studies how human beings can interact with swarm of autonomous robots, is getting more and more attention. Some research focuses on more technical aspects, such as control methods (direct control of the robots, or indirect control of the robots) (*Kolling et al., 2013*), the effect of neglect benevolence (determining the best moment to issue a command to a swarm) (*Walker et al., 2012*; *Nagavalli et al., 2015*), the interaction based on gestures (*Podevijn et al., 2013*; *Nagi et al., 2014*, *Nagi et al., 2015*) or the effect of bandwidth limitation during the interaction (*Nunnally et al., 2012*). These examples do not constitute an exhaustive review of the literature. For a more comprehensive survey, we refer the reader to *Kolling et al. (2016)*.

To date, however, very little research in the human-swarm interaction literature has focused on the psychology of humans interacting with robot swarms. *De la Croix & Egerstedt (2012)* studied the effect of communication network topologies (made by the robots) on humans. The authors found that when humans control a swarm of robots, certain types of topologies increased the workload. *Walker et al. (2013)* and *Amraii et al. (2014)* investigated the effect of two command propagation methods (i.e., methods to propagate a command issued by a human being to all the robots of a swarm) when a human operator guides a leader robot (i.e., a single robot). In their work, a human operator guides the leader robot by changing the leader robot's velocity and heading through a graphical interface. They compared the flooding propagation method to the consensus propagation method. In the flooding propagation method, the robots of the swarm controlled by a human operator all set their velocity and heading to the leader robot's velocity and heading. In the consensus propagation method, the robots of the swarm all set their velocity and heading to the average velocity and heading of their neighbors. The authors showed that the humans' workload is lower in the flooding propagation method than in the consensus propagation method. *Setter et al. (2015)* studied the humans' workload level when a human being guides a robot swarm with an haptic control device (i.e., a device allowing a human to guide the robots and to receive haptic feedback from the robots). *Pendleton & Goodrich (2013)* studied the effect of the robot swarm size (i.e., the number of robots in a swarm) on the human workload level. They conducted an experiment in which participants had to guide swarms of 20, 50 and 100 simulated robots. They found that human workload is not dependent on the number of robots when interacting with a robot swarm. *Podevijn et al. (2016)* studied the

effect of the robot swarm size on the human psychophysiological state. They found that higher robot swarm sizes provoke stronger psychophysiological responses.

With the exception of *Setter et al. (2015)* and *Podevijn et al. (2016)*, all the works that study the psychology of humans interacting with a robot swarm are performed in simulation only. Due to the inherent existence of the reality gap, though, it is not clear if human-swarm interaction studies performed in simulation only would provoke the same psychological reactions as the same human-swarm interaction studies performed with a robot swarm made up of real robots.

The question of the psychological reaction differences when humans interact with a real robot or with a simulated robot has been already addressed in the research field of social robotics. In social robotics, the goal of the robot designers is for the robot to socially interact with humans (*Hegel et al., 2009*). Most of the works that address the question of the humans' psychological reaction differences between the interaction with real robots and simulated robots in social robotics tend to show that humans prefer to interact with a real robot than with a simulated robot. In the following research, all authors used a measure of "enjoyment." The enjoyment is assessed either by a self-developed questionnaire, or by following the game flow model (a model developed to evaluate player enjoyment in games (*Sweetser & Wyeth, 2005*)). When a robot provides humans with help and instructions on a given task, *Kidd & Breazeal (2004)*, *Wainer et al. (2007)* and *Fasola & Matarić (2013)* all reported that humans had a more enjoyable experience (assessed by a self-developed questionnaire) with a real robot compared to a simulated robot. *Pereira et al. (2008)* and *Leite et al. (2008)* also show that humans had a more enjoyable experience with a real robot than with a simulated robot when their participants were playing chess against the robot (both assessed by the game flow model). In *Powers et al. (2007)*, the participants of the authors' study conversed with a real robot and with a simulated robots about health habits. The results of the study revealed that their participants reported to have a more enjoyable conversation with the real robot than with the simulated robot (assessed by a self-developed questionnaire). *Wrobel et al. (2013)* performed an experiment in which elder participants play a card game against a computer, a real robot and a simulated robot. In their results, their participants reported more joy playing against the computer than against the real robot or the simulated robot. However, their participants had a more enjoyable experience playing against the real robot than against the simulated robot (assessed by the game flow model). For a more comprehensive survey about the psychological differences when humans interact with real robots and simulated robots, we refer the reader to *Li (2015)*.

Our work is different from the existing body of research in human-robot interaction because the interaction between humans and robot swarms is inherently different from the interaction between humans and a single robot. This difference is firstly due to the relative simplicity of the robots used in swarm robotics. Robots used in swarm robotics are not equipped with dedicated communication hardware (such as speech-based or text-based communication). Even if they were equipped with dedicated communication hardware, it would be overwhelming—due to the large number of robots— for a human operator to send data (e.g., commands) to and receive data (e.g., feedback)

from each individual robot. A second reason for the difference is that there is no social interaction between human beings and robot swarms.

In this paper, we study the differences in psychological reactions when a human being passively interacts with a real robot swarm, with a simulated robot swarm displayed in a virtual reality environment, and with a simulated robot swarm displayed on a computer screen. Moreover, while all of the aforementioned social robotic works only use dedicated psychological questionnaires to study the participants' psychological reactions, we use a combination of psychological questionnaire and physiological measures in order to study the psychophysiological reactions of participants interacting with a robot swarm.

## METHODOLOGY

### Hypotheses

A review of the human-swarm interaction literature reveals that the majority of the user experiments are performed in simulation. We believe that conducting a human-swarm interaction experiment in simulation can lead to different results than if the same experiment was conducted with real robots. A reason for the results to be different in simulation and in reality is the inherent presence of the reality gap. It is not always possible, however, to perform a human-swarm interaction with real robots (e.g., because an experiment requires a large number of robots). It is our vision that the effects of the reality gap in simulation should be mitigated as much as possible. In order to mitigate the effects of the reality gap, we propose to use virtual reality for simulating the robot swarm. We based the experiment of this paper on these two hypotheses:

- The psychophysiological reactions of humans are stronger when they interact with a real robot swarm than when they interact with a simulated robot swarm.
- The psychophysiological reactions of humans are stronger when they interact with a simulated robot swarm displayed in virtual reality than when they interact with a simulated robot swarm displayed on a computer screen.

Confirming the first hypothesis would imply that human-swarm interaction experiments should be done with real robots instead of simulation. Confirming the second hypothesis would imply that in order to mitigate the effect of the reality gap (if it is not possible to use real robots), it is better for a researcher to simulate a robot swarm in virtual reality because it provokes in humans more realistic psychophysiological reactions compared to simulated robots displayed on a computer screen.

### Experimental scenario

We designed an experimental scenario that allowed us to study the effect of the reality gap on humans in the context of human-swarm interaction. To study the effect of the reality gap, we divided our experimental scenario into three sessions. The order of the three sessions was randomly assigned to our participants. In each session, a participant has to supervise (i.e., watch attentively) a swarm made up of 20 robots. In the so-called *Real Robots* session, the participant supervises a real (i.e., non-simulated) swarm of 20 robots

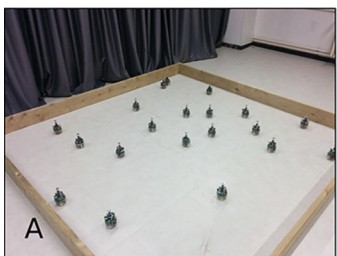 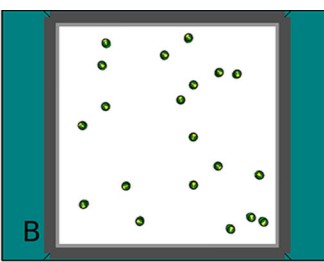 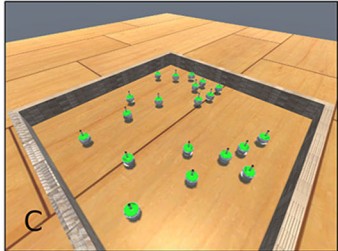

**Figure 2 Robots and environments for each of the three sessions.** (A) View of the real robots and environment. The view is displayed from the participant's perspective. (B) Top view of the robots and of the environment simulated on a computer and displayed on a screen. (C) View of the robots and of the environment simulated in virtual reality. The view is displayed from the participant's perspective.

(see Fig. 2A). In the *Screen Simulation* session, the participant supervises a simulated swarm of 20 robots displayed on a computer screen. In this session, the robot swarm is visible to the participant from the top view (see Fig. 2B). In the *Virtual Reality* session, the participant supervises a simulated swarm of 20 robots displayed in a virtual reality environment. The participant wears a virtual reality headset (i.e., a smartphone put in a Google virtual reality cardboard (https://www.google.com/get/cardboard) and is immersed in a 3D virtual world in which 20 simulated robots are present (see Fig. 2C). During the three sessions (i.e., *Real Robots*, *Screen Simulation*, *Virtual Reality*), the participant has to supervise the robots for a period of 60 s.

## Measures

We used two types of measures: self-reported measures and psychophysiological measures. We use self-reported measures (i.e., data gathered from our participants using a dedicated psychological questionnaire) to determine whether our participants are subjectively conscious of their psychophysiological reaction changes and whether these reaction changes are positive (i.e., our participants report to have a positive experience) or negative (i.e., our participants report to have a negative experience). We use psychophysiological measures, on the other hand, to determine objectively the psychological state of our participants based on physiological responses. These psychophysiological measures are considered objective because it is difficult for humans to intentionally manipulate their physiological responses (for instance to intentionally decrease heart rate). In the following two sections, we first present the self-reported measures used in this study. Then, we present the psychophysiological measures.

### *Self-reported measure*

In this study, we collect our participants' self-reported affective state. We measure our participants' affective state with two scales: valence and arousal. Valence is the cognitive judgement (i.e., pleasure or displeasure) of an evaluation such as the interaction with robots considered in this study. Higher valence values correspond to greater pleasure, while lower valence values correspond to a less pleasurable experience. The arousal scale assesses the mental alertness and the level of physical activity or level of excitation (*Mehrabian, 1996*) felt during an evaluation.

We developed an open source electronic version of the Self-Assessment Manikin (SAM) questionnaire (*Lang, 1980*). This electronic version of the SAM questionnaire runs on a tablet device. The SAM questionnaire represents the values of the arousal scale and of the valence scale with a picture. In this version of the SAM questionnaire, each scale is composed of nine values represented by five pictures and four inter-points between each of the five pictures (i.e., a value of the scale that is not represented by a picture). The tablet application displays the scales in a vertical arrangement where the top-most picture represents the lowest level of the scale (e.g., lowest level of arousal), and the bottom-most picture represents the highest level of the scale (e.g., highest level of arousal). Each picture and each inter-point are associated with a numerical score. Numerical scores vary from 1 to 9. In the valence scale, 1 corresponds to the lowest level of valence (i.e., pleasure is minimal) and 9 corresponds to the highest level of valence (i.e., pleasure is maximal). In the arousal scale, 1 corresponds to the lowest level of arousal (i.e., excitement is minimal) and 9 corresponds to the highest level of arousal (i.e., excitement is maximal). Fig. 3 shows a screen-shot of the SAM questionnaire running on a tablet device.

### Psychophysiological measure

Physiological responses can be used to study the human psychophysiological state (e.g., emotional state or cognitive state). Physiological responses are activated by the autonomic nervous system. The autonomic nervous system is divided into the sympathetic nervous system and the parasympathetic nervous system. The sympathetic nervous system is considered to be responsible for the activation of the fight-or-flight physiological responses (i.e., physiological responses in case of stress). The parasympathetic nervous system, on the other hand, is considered to be responsible for maintaining physiological responses to a normal activity (i.e., the physiological responses at rest).

The electrodermal activity (i.e., the skin's electrical activity) and the cardiovascular activity are two common physiological activities used in the literature to study the autonomic nervous system. In this research, we study our participants' electrodermal activity by monitoring their skin conductance level (SCL) and we study our participants' cardiovascular activity by monitoring their heart rate.

The SCL is a slow variation of the skin conductance over time and is measured in microsiemens ($\mu$S). An increase of the SCL is only due to an increase of the sympathetic nervous system activity. It is, therefore, a measure of choice to study the human fight-or-flight response. SCL has also been correlated to the affective state's arousal (*Boucsein, 2012*). The heart rate is the number of heart beats per unit of time. It is usually measured in beats per minute (BPM). Unlike the SCL though, variation of the heart rate can not be unequivocally associated with a variation of the sympathetic nervous system only. Heart rate can vary due to either a variation of the sympathetic nervous system, a variation of the parasympathetic nervous system, or a combination of both (*Cacioppo, Tassinary & Berntson, 2007*). Heart rate activity is, therefore, more difficult to analyse and interpret than the SCL.

Because the physiological responses can vary between individuals, it is difficult to compare the physiological responses of an individual with those of another. In order

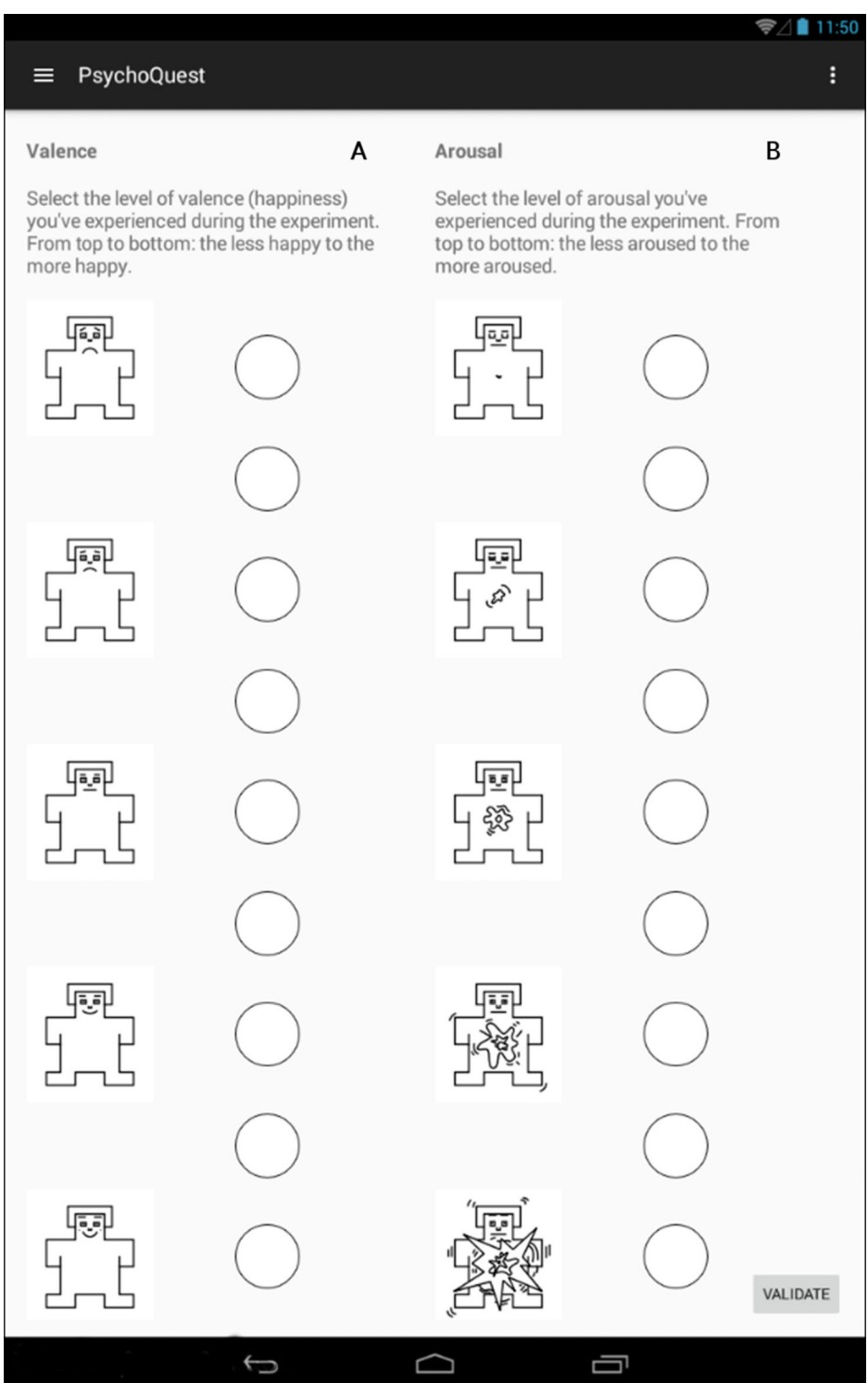

**Figure 3 Electronic version of the Self-Assessment Manikin questionnaire.** (A) the valence scale. The top-most picture corresponds to the lowest level of valence. The bottom-most picture corresponds to the highest level of valence. (B) the arousal scale. The top-most picture corresponds to the lowest level of arousal. The bottom-most picture corresponds to the highest level of arousal. The pictures used in the application are taken from and available at http://www.pxlab.de (last access: April 2016).

to compare the physiological responses between our participants, we first recorded our participants' physiological responses at rest (i.e., the baseline), then we recorded our participants' physiological responses during the experiment. In our statistical analyses, we use the difference between our participants' physiological responses at rest and during the experiment.

## Equipment and experimental setup
### Physiological response acquisition

We monitored our participants' physiological responses with a PowerLab 26T (ADInstruments) data acquisition system augmented with a GSR Amp device. The PowerLab 26T was connected via USB to a laptop computer running Mac OSX Yosemite. We used the software LabChart 8 to record the physiological responses acquired by the PowerLab 26T data acquisition system. We used an infrared photoelectric sensor (i.e., a photopletismograph) to measure the blood volume pulse (BVP) of our participants (i.e., changes in the pulsatile blood flow). The blood volume pulse can be retrieved from the photopletismograph from the peripheral parts of the human body such as on the fingers. We can compute the heart rate from the blood volume pulse. Firstly, we calculate the inter-beat interval (IBI) (i.e., time in seconds between two peaks in the blood volume pulse). Then, we calculate the heart rate by dividing 60 by the IBI. For instance, if the IBI of an individual is 1 s, this individual's heart rate is 60 BPM. Fig. 4A shows the blood volume pulse of a participant during a time window of 10 s. The photopletismograph was attached to the index finger of a participants dominant hand. The photopletismograph was directly connected to the PowerLab 26T.

To monitor the electrodermal activity of our participants, we used brightly polished stainless steel bipolar electrodes connected to the GSR Amp device. These bipolar electrodes were attached to the medial phalanxes of the index and middle fingers of a participants non-dominant hand. In order to monitor the skin conductance, the GSR Amp device applies a direct constant voltage between the bipolar electrodes. The constant voltage is small enough (i.e., 22 mV) to prevent the participants from feeling it. As the voltage is known and constant (22 mV), the GSR Amp device can measure the current between the bipolar electrodes. When the current is known, the GSR Amp device can calculate the conductance of the skin by applying the Ohm's law (conductance is the current measured between the electrodes divided by the constant voltage applied by the GSR Amp device between the electrodes). Fig. 4B shows the skin conductance of a participant during a time window of 10 s.

### Environment and robot behaviour

In all of the three sessions of our experimental scenario (i.e., *Real Robots*, *Virtual Reality*, *Screen Simulation*), we used a square environment of dimension 2 × 2 m. Fig. 2 shows the environment of each of the three sessions. At the beginning of each session, 20 robots are randomly placed in the environment. When an experiment starts, the 20 robots perform a random walk with obstacle avoidance behaviour for a period of 60 s. Each robot executes the two following steps: i) it drives straight with a constant velocity of 10 cm/s,

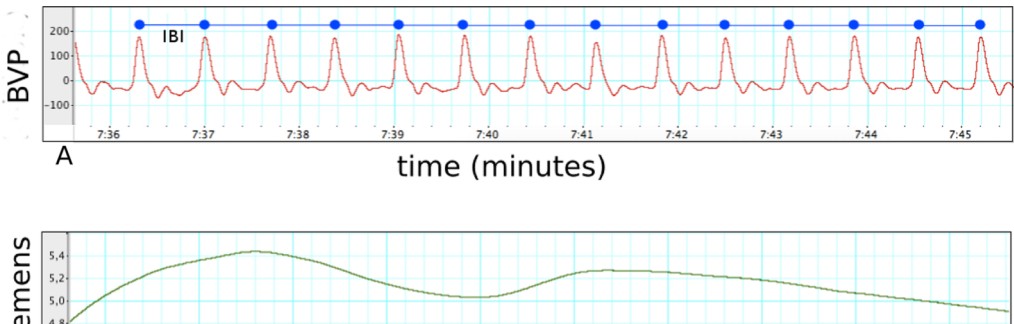

**Figure 4 Physiological measures.** (A) The graph of a participant's blood volume pulse during 10 s. The BVP does not have a standard unit. The x-axis is the time in minutes since the beginning of the recording. The time between two peaks (depicted with two dots connected with a line on the picture) is called the inter-beat interval (IBI). The participant's heart rate (the number of beats per minute) is computed by dividing 60 by the inter-beat interval. In this example, the mean heart rate of the participant during these 10 s is of 87 BPM. (B) The graph of a participant's skin conductance during 10 s. The skin conductance's unit is the microsiemens (y-axis). The x-axis is the time in minutes since the beginning of the recording. The skin conductance is computed by measuring the current flowing between two electrodes and by dividing this current by a constant voltage applied between the electrodes. The skin conductance level of this participant during these 10 s is of 5.17 µS.

and ii) it changes its direction when it encounters either a robot or an obstacle in the direction of movement (i.e., it turns in place until the obstacle is no longer detected in the front part of its chassis).

### Robot platform

The platform used in this study is the wheeled e-puck robot (see Fig. 5) equipped with an extension board. The e-puck robot is designed for educational and research purposes (*Mondada et al., 2009*). The extended version of the e-puck robot is 13 cm high and has a diameter of 7 cm. In this study, we used only a limited subset of the sensors and actuators available to the e-puck robot: the proximity sensors, and the wheel actuators. See *Mondada et al. (2009)* and *Gutiérrez et al. (2008)* for further details and for a complete list of the sensors and actuators available on the e-puck platform. We programmed the e-puck robots using the software infrastructure described in *Garattoni et al. (2015)*.

## Participants

We recruited 28 participants from the campus population of the Université Libre de Bruxelles. All participants were between 18 and 29 years old with an average age of 22.75 years old (*SD* = 3.28). We considered current or anterior cardiovascular problems that could act on the central nervous system as exclusion criteria (i.e., we excluded potential participants with cardiovascular problems). Our participants received an informed consent form explaining that they were filmed during the experiment and that their physiological responses were being collected for research purpose only. At the end of the experiment, we offered a 7€ financial incentive for participation.

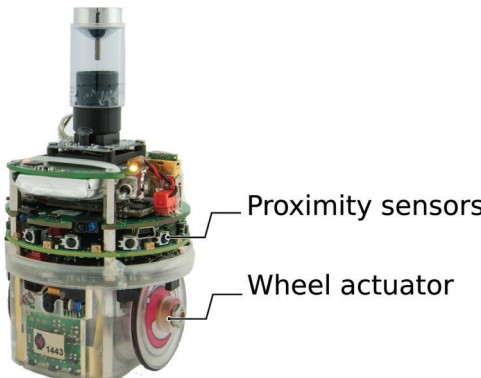

**Figure 5 An e-puck robot used in our experiments.** The proximity sensors are used to detect and avoid nearby robots. The wheel actuators are set to a speed of 10 cm/s.

## Ethics statement

Our participants gave their written informed consent. The experiment was approved by the Ethics Committee of the Faculty of Psychology, Université Libre de Bruxelles (approval number: 061/2015).

## Experimental procedure

We conducted our experiments in the robotic experiment room of the artificial intelligence laboratory at the Université Libre de Bruxelles. Upon arrival, we explained to the participant that she was going to supervise, i.e., watch attentively, a swarm of robots with three different types of visualization interfaces (i.e., on a computer screen, in a virtual reality headset and in reality with real robots). We then showed to the participant the swarm of robots displayed in the three visualization interfaces. The participant was allowed to look at a computer screen displaying a top view of a swarm of robots, to wear the virtual reality headset and to look at the real robots. Once the participant was familiar with the three visualization interfaces, we presented and explained how to answer the electronic version of the SAM questionnaire. Then, we invited the participant to read and sign the consent form. We then asked the participant to wash their hands in clear water (i.e., with no soap) and to remain seated on a chair placed in a corner of the environment used for the *Real Robots* session (see Fig. 1). We attached the participant to two physiological sensors (i.e., a pulse transducer for measuring the participants cardiovascular activity and two finger electrodes for measuring the participants electrodermal activity). We proceeded with a 5 min rest period in order to collect the participant's physiological baseline (i.e., physiological responses at rest). After the 5 min rest period, we started the first session. After each session, we asked the participant to answer the SAM questionnaire. Before starting the next session of the experiment, we collected the participant's baseline during an additional 3 min rest period. This 3 min rest period allowed the participant to get back to a normal physiological activity. During the whole duration of the experiment, the participant remained seated on the same chair. During the *Real Robots* session, the participant was immersed in the environment in which the robots were randomly moving. Prior to the *Virtual Reality* session, we attached

the virtual reality headset to the participant. Prior to the *Screen Simulation* session, we placed a computer screen in front of the participant.

After the experiment ended, we detached the sensors from the participant and conducted a brief interview with her. During the interview, we explained to the participant the goal of the study. Then, we answered our participant's questions. We finished the experiment by thanking the participant and by giving the participant the 7€ incentive. The entire experiments duration was 30 min per participant.

## DATA ANALYSIS AND RESULTS

Out of the 28 participants who took part to the experiment, we had to remove the physiological data (i.e., heart rate and skin conductance) of five participants due to sensor misplacement. We, however, kept the self-reported data (i.e., valence and arousal values reported by the SAM questionnaire) of these five participants. In the following of this section, we analyse the psychophysiological data of 23 participants (15 female and 8 male) and the self-reported data of 28 participants (17 female and 11 male).

We conducted our analyses with the R software (*R Core Team, 2015*) by performing a repeated measures design analysis. Because the data was not normally distributed, we did not use the repeated measure ANOVA test (as the test assumes a normal distribution). Rather, we used a non-parametric Friedman test to analyse both the psychophysiological data and the self-reported data (i.e., the SAM questionnaire). The Friedman test is a rank-based test that does not make any assumption on the distribution of the data. In our case, the Friedman test's null hypothesis is that the medians of the three sessions *Real Robots*, *Virtual Reality* and *Screen Simulation* are the same. In the case of statistical significance of the Friedman test (i.e., at least one session has a different median), we performed a Wilcoxon rank-signed test to evaluate the significance difference between sessions. When the Wilcoxon rank-signed test performed on two sessions is significant, we can conclude that the medians of these two sessions are significantly different. We applied a Bonferroni-Holm correction to the *p*-values returned by the Wilcoxon rank-signed test to take account of the Type I error (i.e., reject the null hypothesis while it is true).

In addition to determining the effect of the reality gap on our participants, we also determined whether psychophysiological data and self-reported data were correlated (e.g., whether skin conductance is correlated with arousal, or whether arousal and valence are correlated). In order to determine this correlation, we performed a Spearman's rank-order correlation test.

We present in Table 1 the results of the psychophysiological and self-reported data (i.e., median and Friedman's mean rank of heart rate, SCL, arousal and valence) in each session (i.e., *Real Robots*, *Virtual Reality*, *Screen Simulation*) as well as the inference statistics of the Friedman tests (i.e., *p*-values and $\chi^2$).

### Psychophysiological data

We did not find any main effect of the reality gap on our participants' heart rate ($\chi^2(2) = 0.78$, $p = 0.67$). However, we found a main effect of the reality gap on our participants' SCL ($\chi^2(2) = 15.2$, $p < 0.001$). Pairwise comparisons on the SCL data

**Table 1 Results of the psychophysiological data and of the self-reported data.** We provide the median and the Friedman's mean rank (in parentheses) of the three sessions (*Real Robots*, *Virtual Reality*, *Screen Simulation*). We also provide the inference statistics of the Friedman test (i.e., $\chi^2$ and $p$).

| Dependent variable | n | Real Robots | Virtual Reality | Screen Simulation | $\chi^2$ | $p$ |
|---|---|---|---|---|---|---|
| Heart rate | 23 | 0.39 (1.87) | 1.01 (2) | 1.44 (2.13) | $\chi^2(2) = 0.78$ | 0.67 |
| SCL | 23 | 4.91 (2.65) | 1.54 (1.78) | 0.77 (1.57) | $\chi^2(2) = 15.2$ | **< 0.001** |
| Arousal | 28 | 6 (2.45) | 5 (2.16) | 3 (1.39) | $\chi^2(2) = 19$ | **< 0.001** |
| Valence | 28 | 7 (2.73) | 6 (1.71) | 5 (1.55) | $\chi^2(2) = 24.87$ | **< 0.001** |

showed a statistical difference between the *Virtual Reality* session and the *Real Robots* session ($Z = 3.9$, $p < 0.001$) and between the *Screen Simulation* session and the *Real Robots* session ($Z = 3.13$, $p < 0.001$) but not between the *Screen Simulation* session and the *Virtual Reality* session ($Z = 0.9$, $p = 0.3$), see Fig. 6B.

### Self-reported data

We found a main effect of the reality gap on our participants' arousal ($\chi^2(2) = 19$, $p < 0.001$) and on our participants' valence ($\chi^2(2) = 24.87$, $p < 0.001$). Pairwise comparisons on the arousal data showed statistical differences between the *Screen Simulation* session and the *Real Robots* session ($Z = 3.97$, $p < 0.001$) and between the *Screen Simulation* session and the *Virtual Reality* session ($Z = 2.55$, $p < 0.05$). There was no statistical difference between the *Virtual Reality* session and the *Real Robots* session ($Z = 1.52$, $p = 0.13$), see Fig. 6C.

Pairwise comparisons on the valence data showed statistical differences between the *Virtual Reality* session and the *Real Robots* session ($Z = 3.6$, $p < 0.001$) and between the *Screen Simulation* session and the *Real Robots* session ($Z = 4$, $p < 0.001$). Pairwise comparisons do not show any statistical difference between the *Screen Simulation* session and the *Virtual Reality* session ($Z = 0.78$, $p = 0.4$), see Fig. 6D.

### Correlations

In addition to studying the effect of the reality gap on our participants, we investigated whether or not some of the dependent variables (i.e., heart rate, skin conductance, arousal and valence) were pair-wise correlated. In order to calculate a correlation between psychophysiological data and self-reported data (e.g., correlation between skin conductance and arousal) we only took into account the self-reported data of the participants whose psychophysiological data had not been rejected (due to sensor misplacement). For the correlation test between arousal and valence we took the 28 participant data points. We did not find any correlation within each of the three sessions (i.e., there was no correlation for any pair-wise dependent variable within the *Real Robots* session nor the *Virtual Reality* session nor the *Screen Simulation* session). We, therefore, investigated whether there was some correlations when the data of each condition was pooled together (e.g., we aggregated skin conductance values from the three sessions). Regarding correlation between psychophysiological data and self-reported data, we found a correlation between skin conductance and valence ($\rho = 0.42$, $p < 0.001$) and a weak correlation between skin conductance and arousal ($\rho = 0.253$, $p = 0.03$).

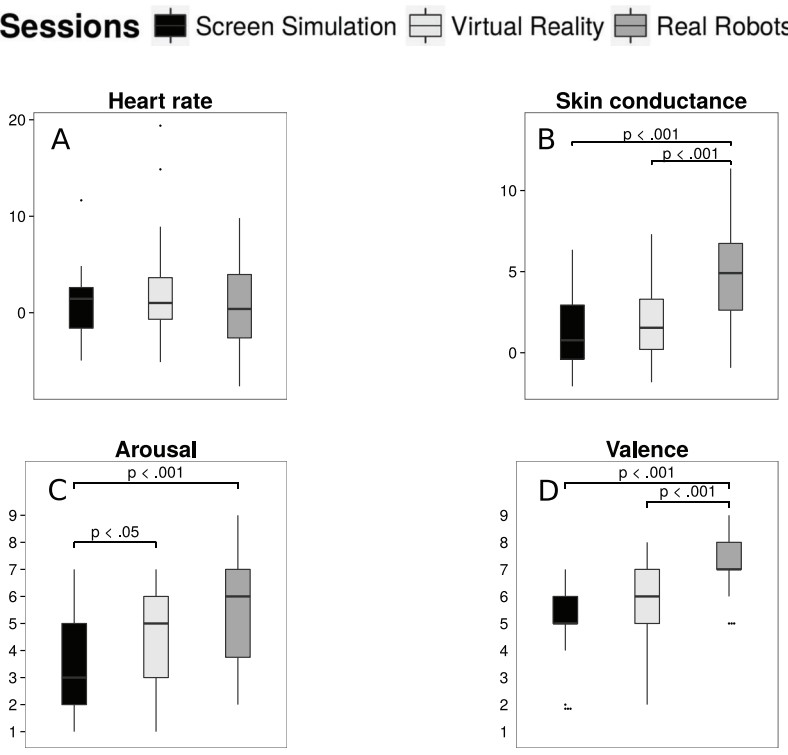

**Figure 6 (A) Boxplots of the heart rate, (B) skin conductance level, (C) arousal and (D) valence.** We visually report the median value of each session with the bold horizontal line. We report the outliers with the dots. Two boxplots are connected when the sessions are significantly different.

There was no correlation between heart rate and valence and between heart rate and arousal. Concerning the self-reported data, we found a correlation between arousal and valence ($\rho = 0.32$, $p = 0.002$). We did not find any correlation between heart rate and skin conductance.

## Gender effect and session order effect

Finally, we also studied the gender effect (i.e., whether females and males differ in their results) and the session order effect (i.e., whether the participants become habituated to the experiment). We analysed the gender effect by splitting into two groups the males' and females' results of each dependent variable (i.e., heart rate, skin conductance, arousal and valence) for each condition (i.e., *Screen Simulation*, *Virtual Reality*, *Real Robots*). We compared these two groups with a Wilcoxon rank-sum test—the equivalent test of the Wilcoxon rank-signed test for independent groups. We did not find any statistically significant difference between males and females in any condition, for any dependent variable. We studied the session order effect as follows. For each condition and for each dependent variable, we separated into three groups the results of the participants who encountered the session first, second or third, respectively. We compared the three groups with a Kruskall-Wallis test—a non-parametric test similar to a Friedman test but for independent groups. We did not find any statistically significant difference among the

three groups in any session, for any dependent variable, suggesting that the session order had no significant effect on our participants.

## DISCUSSION AND CONCLUSION

In this paper, we presented a study on the effect of the reality gap on the psychophysiological reactions of humans interacting with a robot swarm. We had two hypotheses. The first hypothesis stated that humans interacting with a real (i.e., non-simulated) robot swarm have stronger psychophysiological reactions than if they were interacting with a simulated robot swarm. The second hypothesis stated that humans interacting with a simulated robot swarm displayed in a virtual reality environment have stronger psychophysiological reactions than if they were interacting with a simulated robot swarm displayed on a computer screen.

Both the self-reported data (i.e., arousal and valence) and the psychophysiological data (i.e., skin conductance) show that the reality gap has an effect on the human psychophysiological reactions. Our participants had stronger psychophysiological reactions when they were confronted to a real swarm of robots than when they were confronted to a simulated robot swarm (in virtual reality and on a computer screen). These results confirm our first hypothesis.

Of course, it is not always possible for researchers to conduct a human-swarm interaction study with real robots, essentially because real robots are still very expensive for a research lab and real robot experiments are time consuming. It is, therefore, not realistic to expect human-swarm interaction researchers to conduct human-swarm interaction experiments with dozens or hundreds of real robots. For this reason, we decided to investigate the possibility of using virtual reality in order to mitigate the effect of the reality gap. To the best of our knowledge, virtual reality has yet never been used in the research field of human-swarm interaction and is little studied in social robotics (*Li, 2015*). Only the self-reported arousal show that our participants had stronger reactions during simulation in virtual reality than during simulation on a computer screen. With these results, we can not strongly confirm our second hypothesis. However, the results of the skin conductance and the self-reported valence, combined with the significant results of the arousal, both show a trend of our participants to have stronger psychophysiological responses in virtual reality than in front of a computer screen.

In this paper, we designed our experiment based on a purely passive interaction scenario. In a passive interaction scenario, human operators do not issue commands to a robot swarm. We motivated our choice of a passive interaction by the fact that an active interaction could influence the human psychophysiological state (making it difficult to separate the effect of the active interaction and the effect of the reality gap on our participants' psychophysiological state). However, now that we have shown the effect of the reality gap in a purely passive interaction scenario, future work should focus on this effect in an active interaction scenario in which human operators do issue commands to a robot swarm. For instance, we could use the results presented in this paper

as a baseline and compare them with those of an active interaction scenario in which human operators have to guide a swarm in an environment.

In human-swarm interaction, as for any interactive system, it is fundamental to understand the psychological impact of the system on a human operator. To date, in human-swarm interaction research, such understanding is very limited, and worse is often based purely on the study of simulated systems. In this study, we showed that performing a human-swarm interaction study with real robots, compared to simulated robots, significantly changes how humans psychophysiologically react. We, therefore, recommend to use as much as possible real robots for human-swarm interaction research. We also showed that in simulation, a swarm displayed in virtual reality tends to provoke stronger responses than a swarm displayed on a computer screen. These results, therefore, tend to show that if it is not possible for a researcher to use real robots, virtual reality is a better choice than simulation on a computer screen. Even though more research should focus on this statement, we encourage researchers in human-swarm interaction to consider using virtual reality when it is not possible to use a swarm of real robots.

### Funding

This work was partially supported by the European Research Council through the ERC Advanced Grant "E-SWARM: Engineering Swarm Intelligence Systems" (contract 246939). Rehan O'Grady and Marco Dorigo received support from the Belgian F.R.S.–FNRS. The funders had no role in study design, data collection and analysis, decision to publish, or preparation of the manuscript.

### Grant Disclosures

The following grant information was disclosed by the authors:
European Research Council through the ERC Advanced Grant "E-SWARM: Engineering Swarm Intelligence Systems": 246939.

### Competing Interests

Marco Dorigo is an Academic Editor for PeerJ Computer Science.

### Author Contributions

- Gaëtan Podevijn conceived and designed the experiments, performed the experiments, analyzed the data, contributed reagents/materials/analysis tools, wrote the paper, prepared figures and/or tables, performed the computation work, reviewed drafts of the paper.
- Rehan O'Grady conceived and designed the experiments, reviewed drafts of the paper.
- Carole Fantini-Hauwel conceived and designed the experiments, contributed reagents/materials/analysis tools, reviewed drafts of the paper.
- Marco Dorigo conceived and designed the experiments, contributed reagents/materials/analysis tools, reviewed drafts of the paper.

## Ethics

The following information was supplied relating to ethical approvals (i.e., approving body and any reference numbers):

Ethics Committee of the Faculty of Psychology, Université libre de Bruxelles, Approval number: 061/2015.

## Data Deposition

The raw data has been supplied as Supplemental Dataset Files.

## Supplemental Information

Supplemental information for this article can be found online at http://dx.doi.org/ 10.7717/peerj-cs.82#supplemental-information.

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
