# Peer review of "Investigating the effect of the reality gap on the human psychophysiological state in the context of human-swarm interaction"

_PeerJ Computer Science, doi:10.7717/peerj-cs.82_

## Round 0.1 · original submission · Major Revisions

· Academic Editor

Major Revisions

Although all 4 reviewers liked the paper, and agreed that it was well written, reviewers 3 and 4 in particular made suggestions for revision. Please respond to their suggestions and revise accordingly.

Reviewer 1 ·

Basic reporting

No Comments

Experimental design

No Comments

Validity of the findings

No Comments

Comments for the author

The paper shows an interesting topic related to human-being psychophysiological reactions toward activities which have a collaboration between human and machine devices. In reality, there is a big difference of results achieved from simulation environment and reality, especially in matters related to human. Because human has a heterogeneity reaction mechanism even in the same environment, the reactions are dependent on task quantity, under pressure or not, the order of sequence of tasks and so on. Authors indicate that virtual reality devices can be a solution contributes to reducing feeling differences in a comparison between simulation environments and reality. It plays an important role for other study or applicable fields regarding human psychophysiology, but it also would demand another measurement of the impact of virtual reality environment to the awareness of participants.
I am interested in the research topic as well as its application, I suggest to accept the paper.

Reviewer 2 ·

Basic reporting

No comments

Experimental design

No comments

Validity of the findings

No comments

Comments for the author

The paper presents an interesting method on reducing (eliminating) the reality gap on human interactions with robots. I think that it is very important for researchers on virtual (or augmented) reality communities.

Also, the questions has been well designed, and the hypothetical testing has been well done.

·

Basic reporting

This paper describes a set of experiments in which human subjects observe either a "flat screen" simulated robotic swarm, a robotic swarm rendered in a virtual environment, and a real robotic swarm. The aim of the study is to measure the reaction of the subjects to the three different types of swarm using their psychophysiological states, as well as questionnaires.

From my perspective, the paper is extremely well written. The subject is introduced in a very clear way, with reference to relevant literature. My knowledge of this field (human-robot interaction) is limited, but by reading the literature review from this paper, I got a sufficiently clear idea of the state of the art. I can not say whether the authors missed to cite recent important papers.

The paper is structured very well, with clearly stated hypotheses, very good description of methods, results, and conclusions. I really appreciated the justifications for the statistical tests used which significantly help the reader to digest the results. The figures are fine.

Experimental design

I am not used to work with human subjects, so I have no experience with these type of methods. It seems to me that the experimental design is rigorous and solid enough to provide answers to the initial hypothesis. The factor gender is not balanced. Not sure if this matters, probably not, but adding few words on this would be better.
As I have already said, methods, analysis, and results are very well explained.

Validity of the findings

As far as I can say, this is a novel study, which shows new data concerning the human-swarm interaction. Data discussed in the paper looks to me robust, statistically sound, and rigorously controlled.

Comments for the author

I have only minor further comments:
1) It would be nice to say few words on why the interaction with a swarm should be different from the interaction with any other single robot system. If no difference should be made, then your results can be extended to any robotic system, and you could link your study to a broad literature.

2) One part that I found a bit week is about the motivations of the work. The paper, from the title, promise human-swarm interaction. For any expert in swarm robotics, the interest is focused on what the swarm does (or does not). In this work, the actions of the swarm comes after (in term of significance) the response of the subject, which ultimately does not "interact" with the swarm (the subject observes the swarm). I think this secondary role of the behaviour of the swarm, and the non interaction between human observer and swarm has to be made more explicit in the introduction. Otherwise some reader may interpret the "human-swarm interaction" in the title in a wrong way.

3) Point 2 has to be further discussed in the conclusions, when it is time to clarify the relevance of this study for human-swarm interaction. To what extent, an analysis of an observation-based scenario relates to more complex scenarios where the human subjects interact with the swarm? Please, expand this point in the conclusion.

Reviewer 4 ·

Basic reporting

No Comments

Experimental design

No Comments

Validity of the findings

No Comments

Comments for the author

This paper describes an investigation that is timely and interesting. Indeed, the question of whether a simulation can be regarded as equivalent to reality in robotics research has been an object of debate for many years to date. And still one can find nowadays papers accepted in prestigious journals that validate their results in simulation only. This is especially relevant in situations in which the human-robot interaction plays a major role, and this so-called reality gap can make a substantial difference in the psychological effects on the interacting human.

The paper addresses this issue in the context of a recent research field, namely human-swarm interaction. A field in which research in general, and most previous similar studies in particular, are conducted primarily in simulation. For all these reasons, the user study, as described in the paper, is welcome and its hypothesis are relevant and original.

The paper is extremely well-written including all kind of details that make the experiments fully replicable. A strong point of the methodology is the use of both subjective self-reported measures based on questionnaires along with more objective psychophysiological measures (based on skin conductance and heart rate). The experimental procedure is in general correct as well as the statistical analysis of the data. The conclusions are, in principle, reasonably well supported by the experimental results.

However, I have two major concerns regarding the significance of the conclusions.

- The first one refers to the nature of the interaction itself in the experiments. The subjects were instructed to "supervise" the swarm of robots, i.e. "watch attentively" for 60 s. Their behavior was a simple random walk with obstacle avoidance. This watching activity is far from any reciprocal action or influence and, consequently, it is questionable whether this can be considered as an interaction at all. In consequence the validity of their conclusions as pertaining to robot-swarm _interaction_ is limited. Moreover, given the low-level of involvement of the subjects in the robot actions, the expected response of the sympathetic nervous system should be low since this situation will generate little stress in the human. The authors should have made a greater effort to come up with a proper interaction scenario with a greater implication of the participant.

- My second concern is a consequence of the first one and questions the relevance of the study to swarm robotics itself. Since the subjects are essentially watching, the fact that it is robots what they are watching could be psychologically irrelevant, to the extent that they could have been watching non-robotic swarms like a school of fish in a tank, a set of billiard balls moving randomly on a table, or even a complex mechanism with several moving parts with no resemblance to a swarm at all. Would in these cases the results have been the same due to the reality gap? The answer for me is unclear and additional experiments should have been conducted with this kind of conditions as control cases.

Addressing these two questions would require additional experimental work, but I believe it would greatly enhance the merit and impact of research. In this case, the questionnaires should be improved. Asking directly the subjects to select directly their level of valence or arousal is not good practice and there exist much better questionnaires that provide these levels by asking a set of more natural questions about their interaction experience. Probably these questions would not make much sense in the first place since as an interaction experience it was rather poor.

Still the paper could be accepted without additional experimental work if the following recommendations are incorporated:

- The authors write that there is no social interaction between humans and robot swarms. This may be true but they should clarify the notion of human-swarm interaction, describe the nature of the interactions addressed in previous related studies, and justify why their watching action can still be valid as an interaction in this context.

- The authors should discuss the second point above and justify why, under the given circumstances, the study is relevant to swarm robotics and the fact that robots or even swarms were used is not accessory.

- Apparently the order of the three conditions was random for each participant (please clarify this more explicitly). It would be interesting to know whether there are significant differences among the same condition in different order; given the rather passive role of the subjects, the third time they watch the same situation a weaker reaction may result in any case due to habituation.

- In experiments in social HRI results tend to be different depending of the gender of the participants. Having results separated and analyzed by gender would add value to the study.

- When reporting the literature about the reality gap in social HRI the term "more enjoyable" is repeated again and again. The authors should be more precise in their reporting and specify what psychological effect was measured in each case.

- Since the reader is referred to Garattoni et al. (2015) for details about the software infrastructure and it is a non-archival publication, a web link should be provided.

---

## Round 0.2 · accepted · Accept

· Academic Editor

Accept

Thank you for your submission to PeerJ Computer Science. Please check the link as mentioned by Reviewer 4 and provide the correct link while in production.

Reviewer 4 ·

Basic reporting

No comments.

Experimental design

No comments.

Validity of the findings

No comments.

Comments for the author

The authors have convincingly replied to my previous comments and have satisfactorily incorporated my recommendations, including additional analysis regarding the gender effect and that of the order of the conditions.
Regarding my comment about a link to Garattoni et al. (2015), the link provided seems not to work. I got the following error using three browsers. Please check.

The requested URL /IridiaTrSeries/link/IridiaTr2015-004.pdf. was not found on this server.